# Comparison of soil organic carbon and total nitrogen stocks between farmland treated with three and six years level soil bund and adjacent farmland without conservation measure: In the case of southwestern Ethiopia

**Leta Hailu** [1]*, **Mulugeta Betemariyam** [2]

**1** Jimma Agricultural Research Center, Jimma, Ethiopia, **2** Madda Walabu University, Bale Robe, Ethiopia

\* latahailu@gmail.com

## Abstract

This study was conducted to examine and compare the status of soil organic carbon (SOC) and total nitrogen (TN) stocks between farmlands treated with level soil bund (LSB) of three and six years and adjacent farmland without conservation measure (control) at Somodo Watershed. Soil samples were collected from farmland treated with LSB-3 years, LSB-6 years and control using randomized complete block design. A total of 108 composite soil samples (3 treatments * 6 replications * 3 bund zones * 2 depths (0–20 and 20–40 cm) were collected for analysis and determination of the Organic Carbon fraction (OC) and Nitrogen fraction (N). OC was determined using Walkley and Black method while N was determined using the Kjeldahl digestion, distillation and titration method. The result indicated that farmland treated with LSB-6 years has insignificantly higher SOC ($98.43\pm11.55$ Mg ha$^{-1}$) and TN ($9.37\pm1.10$ Mg ha$^{-1}$) stock than control SOC ($93.01\pm13.51$ Mg ha$^{-1}$) and TN ($9.28\pm1.60$ Mg ha$^{-1}$) stock. Likely, farmland treated with LSB-6 years has insignificantly higher SOC and TN stock than farmland treated with LSB-3 years SOC ($96.61\pm11.45$ Mg ha$^{-1}$) stock. With respect to the age of LSB, farmland treated with LSB-6 years accumulated more SOC stock (5.83%) than control. This study revealed that the age of LSB conservation measures has a critical role in enhancing soil fertility through maintaining and sequestering SOC and TN.

## Introduction

Soil erosion is investigated in different disciplines and from different viewpoints. For the purposes of this research, soil erosion is defined as the net long term balance of all factors that detach soil and move it from its original location [1]). Despite different efforts that have been made by different researchers and extensions to mitigate its effects for a century, soil erosion

**Data Availability Statement:** All relevant data are within the manuscript and its Supporting Information files.

**Funding:** Ethiopian Institute of Agricultural Research is the institution that supported this study. The funders had no role in study design, data collection and analysis, decision to publish, or preparation of the manuscript.The corresponding author received a salary from the funders.

**Competing interests:** The authors have declared that no competing interests exist.

by water, wind and tillage continues to be the greatest threat to soil health and soil ecosystem services in many regions of the world [2]. According to [1] estimation, due to the problem of soil erosion, annually about 0.4 percent of global crop yield is reduced.

Soil erosion is the most detrimental ecological process in Ethiopia and degrading the valuable soil resources which are the reservoir of goods and services essential to ecosystems and human well-being. It brings changes in physicochemical properties such as texture, bulk density, infiltration rate, available water, nutrient holding capacity and depth of favorable root growth. These changes have a negative effect on most of the soil ecological function and services [3]. The Ethiopia highland reclamation study reported that, only at the mid-1980, 27 million hectares or almost half of the highland area was significantly eroded and over 2 million hectares were beyond reclamation [4]. Taking losses from both erosion and nutrient depletion [5] estimated a total of 0.5 million tons' crop losses in 1985 at the highlands of Ethiopia. Recent studies also revealed the rates of soil erosion as 20 Mg ha$^{-1}$ year$^{-1}$ on currently cultivated lands and 33 Mg ha$^{-1}$ year$^{-1}$ on formerly cultivated degraded lands in Ethiopia [6]. Lack of an effective watershed management system, low vegetative cover, over grazing and fault crop production and inappropriate soil conservation measures are playing significant role in soil erosion in the Ethiopian highlands [7–9]. In addition, important factors like a slope, aspect and soil types would play a major role in the mechanism of soil erosion [10].

Since the mid-1970s and 80s, different studies conducted in Ethiopia have verified the positive impacts of soil and water conservation practices on soil physicochemical properties and crop yields [11–13]. Among these, mechanical soil water conservation measures (bunds, terraces, check dams, cut off drains and waterways) and biological (homestead and communal tree plantations and enclosures) measures have been implemented in different agroecology of the country [13–15]. For instance, farmland treated by soil bund and stone-faced soil bund structures in the Lole watershed of northwest Ethiopia found significantly improved physicochemical properties of the soils due to the accumulation of fine-textured clay and silt fractions behind the constructed structures [16]. Similarly, conserved farmland exhibited lower bulk density as compared to un-conserved adjacent farmland at Adaa Berga district, western Ethiopia [17]. Moreover, soil and water conservation has also a potential to reduce the loss of runoff and soil by improving water retention capacity on treated farm [18]. The slope gradient treated with soil and water conservation measures for 20 years in Minizr Catchment, Northwest Ethiopia has found a 2.7% slope reduction because of trapped sediment [19].

The Somodo Watershed of southwestern Ethiopia is well known for its inappropriate land use, high population pressure, overgrazing, and erosive tropical rains, which are causing severe soil erosion for the past many years [9,20]. Farmers are practicing farming on a more than the prohibiting range of a slope. This aggravates soil loss and affects the productivity of the agricultural land. Moreover, the eroded sediment also challenging the downstream where, the soil is deposited. To heal the causes of such soil erosion and alleviate the problem, Jimma Agricultural Research Center under the Ethiopian Institute of Agricultural Research have extensively implemented soil and water conservation structures for the past six years. However, no quantitative evidence has been reported on the impacts of LSB on SOC and TN improvement in the watershed. Therefore, the objective of the study was to analyze the status of SOC and TN stocks between farmland treated with LSB of different ages and control. The difference in SOC and TN between the loss zone and accumulation zone within the treated and adjacent untreated plots was also evaluated. The study hypothesized that farmland treated with LSB of different ages has higher SOC and TN than the control due to the availability of retained water for crop growth and return of biomass to the soil as organic matter.

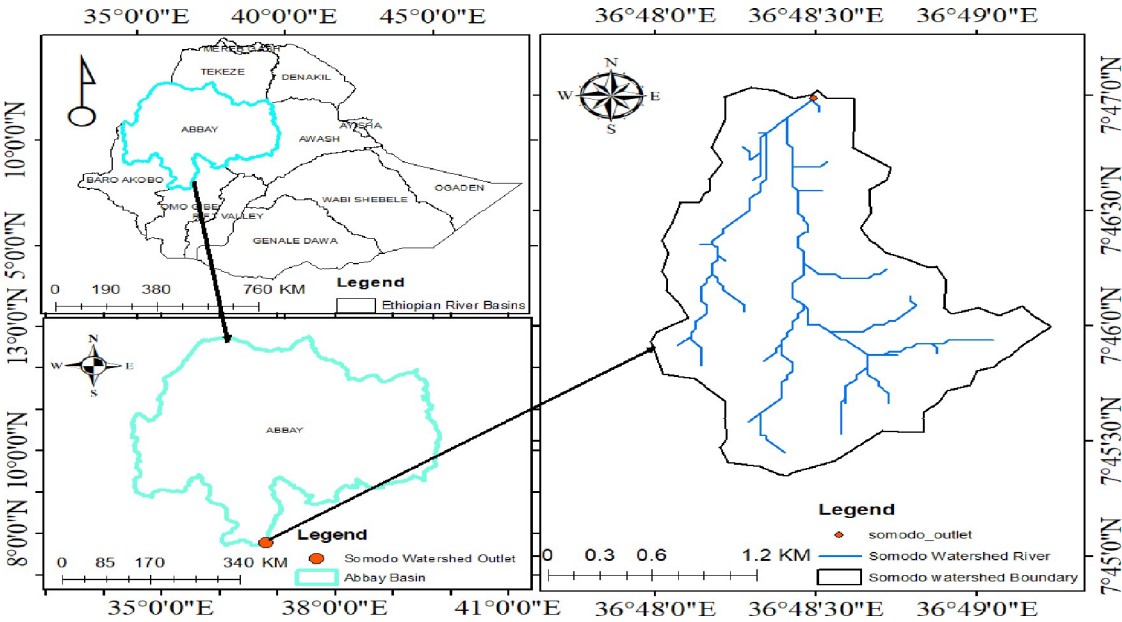

**Fig 1. Map of Somodo watershed, Jimma Zone, South-western of Ethiopia.**

## Materials and methods

### Description of the study area

The study was conducted in the Didessa catchment of Somodo watershed, which is located at about 15 km to West of Jimma Town and 368 km to the South-west of Addis Ababa. Geographically, it is situated between 7˚45"00'N-7˚47"00'N latitude and 36˚48"00'E-36˚49"00'E longitude (Fig 1). It covers an area of 400 ha with an altitude ranging from 1900 to 2075 m above sea level.

The area has a bimodal rainfall distribution with maximum rainfall between July and September and moderate rainfall between March and May. The long-time means annual rainfall (16 years) of the watershed is 1948.0 mm. The means monthly temperature of the site is 19.27˚C ranging from 13.6˚C and 25˚C (Fig 2). Nitisols is the most dominant soil type and sandy clay loam is the textural soil class of the watershed [21]. Nitisols are well drained, red or reddish soils with diffuse horizon boundaries and a nitic horizon with more than 30% clay [22]. The major land use system of the watershed was agricultural land, forest land, grazing land and Agroforestry practices [19].

### Research design and soil sampling

Soil samples were collected from farmland treated with LSB-3 years, LSB-6 years and adjacent control using randomized complete block design. The samples were collected randomly from the fields by establishing 10*10 m plot for each treatment within a similar range of altitude and location used as a block to reduce soil property variation due to micro-topographic differences.

The composite and undisturbed soil samples were collected from each consecutive inter-bund zones (upper, middle and lower) at two depths (0-20cm and 20-40cm) using an auger and a core sampler respectively, assuming that the deposited sediment depth due to the implemented LSB will not exceed this depth. A total of 108 composite soil samples (3 treatments * 6

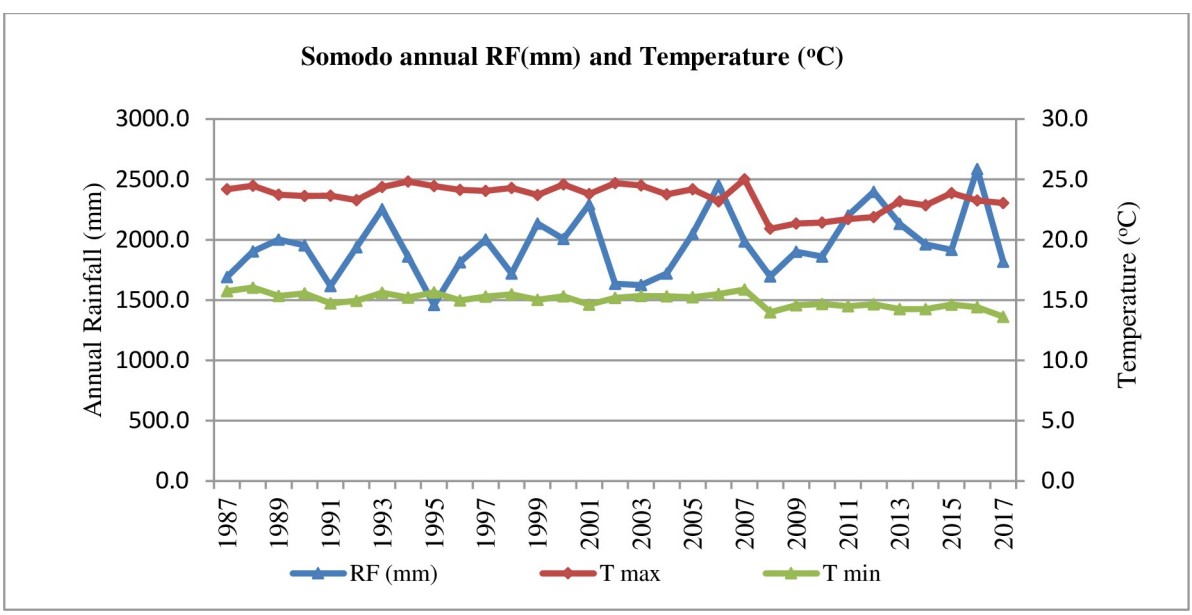

**Fig 2. Mean of annual rain fall (mm) and temperature (˚C) of Sodomo watershed.**

replications * 3 bund zones * 2 depths (0–20 and 20–40 cm) were collected for OC and N analysis and for bulk density determination.

## Laboratory analyses

The collected soil samples were transported to Jimma Agricultural Research Center laboratory and Jimma, University College of Agriculture and Veterinary medicine for analysis and determination of OC, N and bulk density. Bulk density was determined by the core method, and samples are dried in an oven at 105˚C for 48 hours [23]. OC was determined by Walkley and Black method [24]. N was determined using the Kjeldahl digestion, distillation and titration method [25]. Finally, SOC (Mg ha$^{-1}$) and TN (Mg ha$^{-1}$) stock were calculated based on the fixed depth (FD) approach which is expressed as the product of respective carbon and nitrogen fraction (%), bulk density (g/cm$^3$), and layer thickness (cm).

## Statistical analysis

All data were checked for normality prior to doing the analysis of variance using the Kolmogorov-Smirnov test. Statistical differences in the SOC and TN among treatments (with and without LSB), bund zones, and bund ages in the top 40 cm of soil depth were tested following the general linear model (GLM) procedure of SPSS Version 20.0 for Windows. Pearson correlation was also performed to estimate the correlation relationship between SOC and TN within the watershed.

## Results and discussion

### SOC and TN stock comparison between farmland treated with LSB-3 years and control

The SOC stock of farmland treated with LSB-3 years and control did not showed a significant difference. However, the higher mean value of SOC stock was observed in the farmland treated with LSB-3 years (96.61±11.45 Mg ha$^{-1}$) when compared with the SOC stock

**Table 1. (Mean ±Std. Deviation) of SOC (Mg ha$^{-1}$) and TN (Mg ha$^{-1}$) stock of farmland treated with LSB-3 years, LSB-6 years and control.**

| Bund Zone | Depth | SOC stock | | | TN stock | | |
|---|---|---|---|---|---|---|---|
| | | Control | LSB-3 years | LSB-6 years | Control | LSB-3 years | LSB-6 years |
| Upper | 0–20 | 45.38±5.84 | 54.72±6.55 | 51.34±6.30 | 5.23±0.77 | 4.50±0.88 | 4.96±0.72 |
| | 20–40 | 39.19±5.34 | 46.94±6.50 | 46.34±9.52 | 4.75±1.45 | 4.33±0.49 | 4.57±0.28 |
| | 0–40 | 84.57±9.22[a] | 101.66±11.96[b] | 97.67±15.43[b] | 9.98±1.93[a] | 8.83±1.36[a] | 9.53±0.63[a] |
| Middle | 0–20 | 50.35±7.67 | 47.74±7.42 | 51.89±5.76 | 4.59±1.40 | 4.36±1.21 | 4.79±1.42 |
| | 20–40 | 43.29±7.69 | 46.06±8.435 | 47.93±5.37 | 4.71±1.13 | 5.12±1.27 | 4.69±0.69 |
| | 0–40 | 93.64±10.27[a] | 93.80±14.08[a] | 99.82±10.11[a] | 9.30±1.35[a] | 9.48±2.06[a] | 9.48±1.86[a] |
| Lower | 0–20 | 53.66±9.29 | 51.55±4.99 | 52.30±4.91 | 4.26±0.53 | 5.25±0.62 | 4.89±0.28 |
| | 20–40 | 47.16±8.24 | 42.82±3.00 | 45.51±6.25 | 4.30±1.25 | 4.75±1.00 | 4.20±0.31 |
| | 0–40 | 100.82±16.46[a] | 94.37±7.65[a] | 97.81±10.49[a] | 8.56±1.37[a] | 10.00±1.22[a] | 9.10±0.37[a] |
| Total | 0–20 | 49.80±8.06 | 51.33±6.67 | 51.84±5.35 | 4.69±1.00 | 4.70±0.97 | 4.88±0.88 |
| | 20–40 | 43.21±7.55 | 45.28±6.27 | 46.59±6.91 | 4.59±1.22 | 4.73±0.97 | 4.49±0.49 |
| | 0–40 | 93.01±13.51[a] | 96.61±11.45[a] | 98.43±11.55[a] | 9.28±1.60[a] | 9.44±1.57[a] | 9.37±1.10[a] |

Means followed by the same letter(s) horizontally for the same parameter are not significantly different at (p ≤ 0.05) with respect to treatments and bund zones.

(93.01±13.51 Mg ha$^{-1}$) of control (Table 1). Regarding the inter bund zones, the minimum, medium and maximum SOC stock for the control were recorded in the upper, middle and lower inter bund zones, respectively. While for farmland treated with LSB-3 years, the maximum SOC stock was recorded at the upper bund zone when relatively compared with the lower bund zone. The upper bund zone of farmland treated with LSB-3 years had significantly higher mean value of the SOC stock (101.66±11.96 Mg ha$^{-1}$) as compared with the adjacent upper bund zone of control (84.57±9.22 Mg C ha$^{-1}$). This result confirms that LSB-3 years can keep the soil at the upper bund zone (loss area) in place and reduce both the on-site and off-site effects of soil erosion. However, the SOC stocks at middle and lower inter bund zones of farmland treated with LSB-3 years and control did not a show significant difference. The non-significant difference in SOC at these bund zones probably due to the young age of LSB and level of past erosion before the structure was built.

In line with this case study, different studies [17,26] have also reported that fields treated with above three years' soil and water conservation measures had higher organic matter accumulation as compared to the non-conserved fields. But the research report on level soil bund without any agronomic or biological techniques by [27] showed no improvement in soil fertility during the first 3–5 years that did not agree with the result of this study. Concerning to soil depth, within each farmland and bund zone, SOC stock was higher (p>0.05) in the top layer (0–20 cm) than in the lower layer (20–40 cm) but the mean difference was not significant (p>0.05) (Fig 3). This find is in line with the finding of [28] that showed the decrement of SOC concentration with the increment of soil depth.

The overall mean TN stock (0–40 cm depth) was not significantly different between farmland treated with LSB-3 years and control (Table 1). But, higher TN stock (9.44±1.57 Mg ha$^{-1}$) was observed in the farmland treated with LSB-3 years than TN stock (9.28±1.60 Mg ha$^{-1}$) of control. This low TN in control might be due to the unrestricted downward movement of organic matter with runoff water from the upper landscape. According to [2], organic matter accumulation is often favored at the bottom of hills due to its transport through erosion from the upper and mid position to the lowest point in the landscape. With respect to soil depth, except the middle bund zone, the TN stocks was higher in the upper layer (0–20 cm) than in the lower layer (20–40 cm) for both farmlands treated with LSB-3 years and control (Table 1).

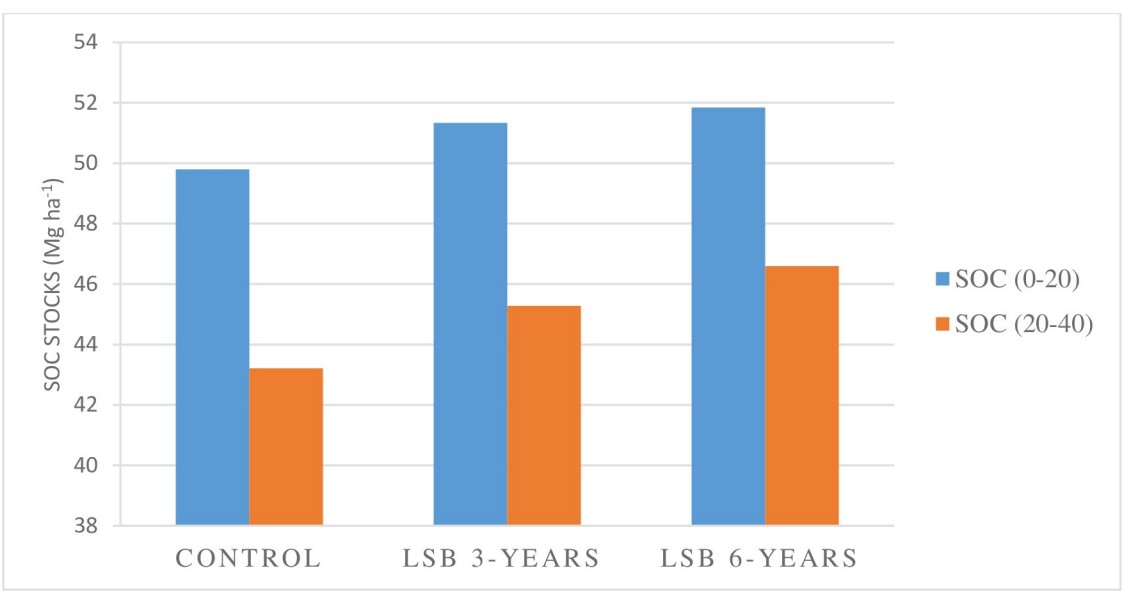

**Fig 3. Status of SOC in LSB 3-years LSB 6 years treated and untreated farmland with respect with soil depth.**

## SOC and TN comparison between farmland treated with LSB-6 years and control

Overall, the mean of SOC stock did not exhibit a significant (P>0.05) variation between farmland treated with LSB-6 years and control (Table 1). Concerning zones, the upper bund zone (97.67±15.43 Mg ha$^{-1}$) of farmland treated with LSB-6 years was significantly higher than the respective adjacent upper bund zone of control (84.57±9.22 Mg ha$^{-1}$) (Table 1). However, variations in SOC stocks among the middle and lower bund zones of farmland treated with LSB-6 years and adjacent control were not statistically significant (P>0.05).

This finding indicates that long term age of the LSB (greater than six years) has a key role for significant change of SOC in the middle and lower bund zones of treated farmland than the corresponding untreated bund zones. Many research results also confirmed that soil erosion resulted in nutrient depletion through reducing and changing the physicochemical conditions of the soil like soil organic matter content, soil structure, water holding capacity, soil bulk density, soil porosity, soil pH and its workability [9,29,30]. The estimated overall mean (0–40 cm depth) SOC stocks for middle and lower bund zone of farmland treated with LSB-6 years was 99.82±10.11 Mg ha$^{-1}$ and 97.81±10.49 Mg ha$^{-1}$, respectively.

TN stock showed no significant variation between farmlands treated with LSB-6 years and adjacent control. However, the overall mean value of TN in farmlands treated with LSB- 6 years (9.37±1.10 Mg ha$^{-1}$) was relatively higher than the TN stock (9.28±1.60 Mg ha$^{-1}$) of control (Table 1). Except for the middle one, within each bund zone, TN stock was higher in the top layer (0–20 cm) than in the lower layer (20–40 cm). The lower mean value of TN observed in the control might be allied with the removal of fertile topsoil by erosion process and the use of crop residues for fuel and animal feed rather than leaving in the farm to decompose and enrich the soil organic matter content. Different case studies showed that TN content of the soil is directly associated with the amount of organic matter constituted in the soil [17,31]. So, if organic matter input from crop residues, manure and any other sources were not balanced the rate of decomposition, there is a faster TN depletion.

### SOC and TN comparison between farmland treated with LSB-3 years and LSB-6 years

The SOC stock did not show a significant difference (P>0.05) between farmland treated with LSB-3 years and LSB-6 years. However, LSB-6 year exhibited more SOC stock at the upper, middle and lower bund zone (Table 1). Similar study on LSB in Bokole watershed, Dawuro zone, Southern Ethiopia also showed that SOC in treated land with LSB-6 years was insignificantly higher than the treated land with LSB-4 years [31].

TN socks of LSB-3 and LSB-6 treated farmlands did not show significant difference (p> 0.05). However, LSB-3 years showed higher overall mean value of TN stock (9.44±1.57 Mg ha$^{-1}$) than LSB-6 years (9.37±1.10 Mg ha$^{-1}$) (Table 1). This might be due to the variation in application of nitrogen containing inputs such as commercial fertilizer, plant residues and animal manure on the farmland before LSB intervention. [32] also reported as the increment of TN stock of treated farmland is determined by the past deposition of soil materials and physical, chemical, biological and anthropogenic factors with complex interactions.

As it was reported in most of the studies [17,33,34], in this study there was a positive and significant correlation between SOC and TN ($R^2 = 0.86$). This confirms the contribution of enhanced organic matter resulted from the implementation of LSB structure stabilized with Vetiver grass as a significant role to increase TN.

## Conclusions

This study confirmed that the proper construction of LSB structure has been influenced the accumulation of the SOC and TN of treated fields as compared to adjacent land without conservation measures. However, the effective outcome is basically based on the age of the structure. That means; old aged treated farmland has enhanced higher SOC and TN than young aged treated farmlands. With respect to inter bund zones, higher SOC and TN were accumulated in the lower bund zone than the upper and middle inter bund zones. This is due to the downward movement of organic matter with runoff water from the upper zone and accumulation of it at the lower area. Since this study is solely focused on the analysis and comparison of SOC and TN effects of LSB without incorporating some other physicochemical properties, socio-economic, institutional and physical aspects of conservation approach due to various limitations, further researches are required to get a comprehensive conclusion. Finally, it could be concluded that; LSB-3 years and LSB-6 years accumulates more amounts of SOC and TN than adjacent non-treated farmlands and hence this contributes to mitigating climate change by preventing erosion-induced greenhouse gas emission into the atmosphere.

## Supporting information

**S1 Appendix. Processed data of SOC stocks.**
(XLSX)

**S2 Appendix. Processed data of TN stocks.**
(XLSX)

## Acknowledgments

We acknowledge the logistic and technical support we got from Jimma Agricultural Research Center and Jimma University college of Agriculture and Veterinary Medicine. We are also thankful to the farmers of Somodo Watershed, who kindly allowed us to take measurements

on their farmlands. Our gratitude goes to Mr. Frew Kapito, Tamirat Kebede and Murad Abaraya, for their ample support on field-work that makes our stay very fruitful.

## Author Contributions

**Conceptualization:** Leta Hailu.

**Data curation:** Leta Hailu.

**Formal analysis:** Leta Hailu, Mulugeta Betemariyam.

**Methodology:** Leta Hailu.

**Resources:** Leta Hailu.

**Software:** Mulugeta Betemariyam.

**Writing – original draft:** Leta Hailu.

**Writing – review & editing:** Leta Hailu, Mulugeta Betemariyam.

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
