## [Decision Letter · Decision Letter 0]

20 Jan 2021

PONE-D-20-35754

Comparison of soil organic carbon and total nitrogen stocks between farmland treated with level soil bund of different age and adjacent farmland without conservation measure: in the case of southwestern Ethiopia

PLOS ONE

Dear Dr. Hailu,

Thank you for submitting your manuscript to PLOS ONE. After careful consideration, we feel that it has merit but does not fully meet PLOS ONE’s publication criteria as it currently stands. Therefore, we invite you to submit a revised version of the manuscript that addresses the points raised during the review process.

We look forward to receiving your revised manuscript.

Kind regards,

Sergio Saia, Ph.D.

Academic Editor

PLOS ONE

Journal Requirements:

2.Thank you for stating the following financial disclosure:

 "No"

3.Thank you for stating the following in your Competing Interests section: 

"No"

4.In your Data Availability statement, you have not specified where the minimal data set underlying the results described in your manuscript can be found. PLOS defines a study's minimal data set as the underlying data used to reach the conclusions drawn in the manuscript and any additional data required to replicate the reported study findings in their entirety. All PLOS journals require that the minimal data set be made fully available. For more information about our data policy, please see http://journals.plos.org/plosone/s/data-availability.

6. Please amend either the abstract on the online submission form (via Edit Submission) or the abstract in the manuscript so that they are identical.

7. We note that [Figure(s) 1] in your submission contain map images which may be copyrighted. All PLOS content is published under the Creative Commons Attribution License (CC BY 4.0), which means that the manuscript, images, and Supporting Information files will be freely available online, and any third party is permitted to access, download, copy, distribute, and use these materials in any way, even commercially, with proper attribution. For these reasons, we cannot publish previously copyrighted maps or satellite images created using proprietary data, such as Google software (Google Maps, Street View, and Earth). For more information, see our copyright guidelines: http://journals.plos.org/plosone/s/licenses-and-copyright.

1.    You may seek permission from the original copyright holder of Figure(s) [1] to publish the content specifically under the CC BY 4.0 license. 

9.We noticed you have some minor occurrence of overlapping text with the following previous publication(s), which needs to be addressed:

http://article.sciencepublishinggroup.com/html/10.11648.j.ijnrem.20160102.15.html

https://file.scirp.org/Html/6776.html

https://www.gssrr.org/index.php/JournalOfBasicAndApplied/article/view/6167

In your revision ensure you cite all your sources (including your own works), and quote or rephrase any duplicated text outside the methods section. Further consideration is dependent on these concerns being addressed.

Additional Editor Comments:

Specifically, it seems that your manuscript reports similar results of those reported by the work indicated from 2 out of 3 reviewers: Leta Hailu, Fantaw Yimer, Teklu Erkossa, 2020. Evaluation of the effectiveness of level soil bund and soil bund age on selected soil physicochemical properties in Somodo Watershed, Jimma Zone, SouthWestern Ethiopia J. Degrade. Min. Land Manage., pp. 2491-2502 DOI https://doi.org/10.15243/jdmlm.2020.081.2491

In the article in "Journal of Degraded and Mining Lands Management" wide parts of the abstract are identical to the present ms. In the cover letter, you specified <<I confirm that this work is original and has not been published elsewhere, nor is it currently under consideration for publication elsewhere>>. I wish to pinpoint that Plos1 consider the acceptance of a ms as a research article without taking into account its novelty and potential impact, upon that the methods are correct. However, Plos1 does not allow to publish data already published in other journals and the present data seems to derive from another articles, with a computation of stock.

In the Hailu et al. J. Degrade. Min. Land Manage article, you seem to show the data on SOC and STN concentrations (and other soil traits) at varying the age of soil bund (0, 3, 6 years), whereas in the present ms in Plos1 you show the SOC and STN stocks at varying the time of soil bund (3 or 6 year, in tab 1 and 2, respectively) while varying the presence of not of conservative management practices in apparently similar zones and layers than the previous. Data on the farms without the conservation measure in tab 1 and 2 are the same and may be derived from the data at 0 years of soil bunds. Lastly, data of tab. 3 are the same of those shown in tab 1 and 2 relative to the conservation measures.

Also, it sounds quite strange to me that here in this ms you don’t cite the former article, that was published before the submission of the present ms, whom results would help you discussing these ones.

It seems thus that the data on a stock basis in the present manuscript may derive from the ones of the former article, thus invite you to revise the present ms and clearly indicate the differences between the article in the J. Degrade. Min. Land Manage and the present one. Also, please carefully take into account the reviewer’s suggestions.

Lastly, I invite you to clearly diversify the previous article from the present ms. Wide parts of the materials and methods may be shared, but not of the other sections. In addition, data should not be replicated within the ms, so that just 1 table is enough.

In addition, tab. 4 is useless. You can just report the one number (R=0.86) in the main text.

Finally, if the data on the Stocks you are reporting derive from the previous work, this should be spelled out and cited in the abstract of the present work, given that stock directly derive from a computation from the concentration, bulk density and depth, all of which can be found in your previous ms. Finally, please indicate if the computation of the stock take into account of the coarse fraction, if any.

Please mind that decision to allow you to opt for a revision is not a pre-requisite for the acceptance of the ms as a research article.  

Reviewers' comments:

Reviewer's Responses to Questions

**Comments to the Author**

1. Is the manuscript technically sound, and do the data support the conclusions?

Reviewer #1: Yes

Reviewer #2: Partly

Reviewer #3: Partly

2. Has the statistical analysis been performed appropriately and rigorously? 

Reviewer #1: Yes

Reviewer #2: I Don't Know

Reviewer #3: Yes

3. Have the authors made all data underlying the findings in their manuscript fully available?

Reviewer #1: No

Reviewer #2: No

Reviewer #3: No

4. Is the manuscript presented in an intelligible fashion and written in standard English?

Reviewer #1: Yes

Reviewer #2: No

Reviewer #3: Yes

5. Review Comments to the Author

Reviewer #1: Review report

Manuscript Number: PONE-D-20-35754 “Comparison of soil organic carbon and total nitrogen stocks between farmland treated with level soil bund of different age and adjacent farmland without conservation measure: in the case of southwestern Ethiopia”

The manuscript addresses a subject of current interest related to the effect of level soil bund (aged 3 and 6 years) an) on Soil organic carbon and total nitrogen Stocks compared to a control farmland. Overall, I do like this paper, but the novelty and potential for impact are somewhat limited. I here encourage the authors to revise the paper.

Some suggestions for further improvement are given below:

1. The paper is written in poor English. There are many grammar mistakes and some sentences that are not easily understood. A very careful and intensive revision by a native speaker is necessary.

2. The title corresponds to the text. However, it has to be redone and reformulated.

3. The authors must pay attention to the abbreviation. Try to use the same abbreviation. Please specify if the values are for SOC or TN amount or for the stocks .

Abstract section

4. Line 22 : “The result indicated that farmland treated with LSB of 6 years has insignificantly higher SOC (102.57±9.00 Mg ha-1) and TN (9.74±0.9 Mg ha-1) than farmland treated with LSB of 3 years (98.42±10.24 Mg ha-1).” It is preferable that the authors add the value of TN for LSB of 3 years for a better comprehension.

5. Line 26 :“Similarly, farmland treated with LSB of 6 years was sequestered more 5.20% % of SOC ..” Please delete the double symbol %.

Introduction section

6. The introduction should be developed. , there are many researches related to this problematic have been done. It would benefit of more detail and supported with the related references.

7. Line 48 :“In addition, important terrain like a slope, aspect and soil types...” Slope, aspect and soil types are considered like factors or parameters not terrain.

8. Line 58 “ Similarly, fields treated with soil and water conservation practices were founded the lower mean bulk density than the untreated fields in Adaa Berga district, western Ethiopia [15].” Please rephrase appropriately

Materials and methods section

9. Description of the study area, line 86 :“ 7°46’’00’N-7°47’’00’N latitude and 36°44’’10’E-36°46’’50’E longitude (Fig 1).” These coordinates system does not correspond to the Fig 1.

10. Figure 2 : On the axis of temperature please add the unit and delete “Max. And Min.” because it is defined on the legend.

11. Research design and soil sampling, line 103-106 and line 107-112 : Please rephrase appropriately to avoid redundancy.

12. Research design and soil sampling, line 115 “Finally, SOC (Mg ha-1) and TN (Mg ha-1) were calculated….” Please specify that it is SOC and TN stocks.

13. Research design and soil sampling, line 112-117 “The collected soil samples were transported to Jimma Agricultural Research Centre laboratory … bulk density (g/cm3), and layer thickness (cm).” The authors should move this paragraph in the laboratory analysis section.

14. The authors must improve the table 1, 2 and 3 : Specify that the values are for SOC and TN stocks as l mentioned before. Also, please try to be brief when writing the name of x-axis, it could be SOC stock LSB-3 years / LSB-6years/ Control.

Results and discussion section

15. The authors showed that total nitrogen, organic carbon and bulk density were used in this paper, but in the results and discussion section, why it was not included? The authors should add these data to this section using a table.

16. Line 149 : TOC and TN data can not completely represent soil organic matter content. As fields treated with above three years’ soil and water conservation measures had higher organic matter accumulation as compared to the non-conserved fields. The data of OM would be more accurate.

17. Figure 3: Please specify that the values are for SOC stocks, on the y-axis and for the legend. Same as I mentioned below, try to write briefly, instead of writing “farmland with three years LSB” The authors can just write LSB-3years and Control.

18. Line 202-204 :“The top layer accounted for 52% of the TN in the upper bund zone of farmland treated with six years aged LSB and 51% of the TN in adjacent upper zone of farmland without conservation measures.” Please provide explanation how the authors get these values?

19. Line 228 “The upper, middle and lower bund zones of SOC stocks in farmland treated with six years LSB was approximately 10.5%, 10.3% and 10.3% higher than the upper, middle and lower bund zones of farmland treated with three years LSB, respectively.” Please explain how the authors get these values?

20. For table 4: The Pearson`s correlation matrix for soil organic carbon and total Nitrogen, is it for the stocks values ?

21. References: Please be sure that all the references cited in the manuscript are also included in the reference list and vice versa with matching spellings and dates and according to the journal format, "Instructions for Authors".

Looking forward for your positive consideration of these comments.

Regards,

Reviewer of the manuscript

Reviewer #2: The Hailu & Betemariyam manuscript compares organic carbon and total nitrogen stocks as critically important soil quality parameters in farmland treated with 3 and 6 years aged level soil bund and adjacent farmland without conservation measures. The results section is divided into three subsections, but due to a data repetition in each subsection please consider a possibility to construct the results section comparing all 3 treatments in one table, if it`s possible to present data of statistical analysis this way.

Data Availability Statement in the manuscript PDF file is Yes, but there is no information where the data may be found.

Add information to Research design section about the soil you were studied: Was it Nitisol or Acrisol?, What is the clay content in it?, etc. Is a sampling site on adjacent farmland without conservation measures representative? Was it the same soil type? Which type of crops were grown on adjacent farmland?

Consider adding initial data (bulk density and content of organic carbon and total nitrogen in the soil) to the results section. This will give additional information to understand the reasons for your findings.

Other than that there were the following minor observations:

Laboratory analyses. Line 120: Walkley and black (use Upper case)

P. 7 Line 172: Text is about TN stocks but the reference is to a figure about SOC stocks (Fig. 3). Delete this reference and insert Figure 3 after Line 154.

Conclusions. Line 261: Finally, it could be concluded that… Avoid “enormous amounts of SOC and TN” because as you`ve found the mean difference in SOC and TN stocks is not significant between 3-years aged LSB and non-treated farmland. To conclude “considerable role in mitigating climate change by sequestering corresponding greenhouse gases” research findings do not seem to be completely enough.

Fig. 3: Delete “a” and “b” symbols in the figure?

Fig. 3 and tables: Round off numbers to 2 decimal places

Consider language editing to make some sentences more easily understandable

Reviewer #3: Dear Authors, unfortunately, the manuscript seems very similar to a previous research published form the same or almost the same Authors. I cannot discuss about the quality if a similar experiment is already published "" ext-link-type="uri" xlink:type="simple">https://jdmlm.ub.ac.id/index.php/jdmlm/article/view/751".

I would be pleased to be invited to revise other research of yours, the topic is sound and actual.

Please try to provide a systematic literature search before to do any experiment, it might be useful perform a more comprehensive search with a structured query in a citation and abstract database such as Scopus or Web of Knowledge, as it was carried out in Modelling of Soil Organic Carbon in the Mediterranean area: a systematic map November 2018Rendiconti Online Societa Geologica Italiana 46/2018 DOI: 10.3301/ROL.2018.68

I suggest to read also other systematic map applications to improve the state of the art writing style.

Kind regards

6. PLOS authors have the option to publish the peer review history of their article (what does this mean?). If published, this will include your full peer review and any attached files.

Reviewer #1: No

Reviewer #2: No

Reviewer #3: No

---

## [Author Response · Author response to Decision Letter 0]

3 Mar 2021

Dear Editor-in Chief 

We are very grateful to the anonymous reviewers for their valuable and constructive comments to our manuscript entitled “Comparison of soil organic carbon and total nitrogen stocks between farmland treated with level soil bund and adjacent farmland without conservation measure: in the case of southwestern Ethiopia, ms number “PONE-D-20-35754”. We have endeavored to address all comments in this revised manuscript and believe the manuscript is now greatly improved, including the language. We highlighted yellow colour on the “revised Manuscript with Track Change” for highlights additional inputs and changes made to the original version. The page where specific responses given for comments are highlighted red colour in Authors’ responses page. Below are our responses to the specific comments raised by the reviewers.

Queries related to manuscript format and funding source 

We ensured that the style of our manuscript meet the PLOS ONE style requirements. 

2a. please clarify the sources of funding (financial or material support) for your study. List the grants or organizations that supported your study, including funding received from your institution.

Ethiopian Institute of Agricultural Research” is the institution that supported this study 

2b. State what role the funders took in the study. If the funders had no role in your study, please state: “The funders had no role in study design, data collection and analysis, decision to publish, or preparation of the manuscript.” 

The funders had no role in study design, data collection and analysis, decision to publish, or preparation of the manuscript 

2c. if any authors received a salary from any of your funders, please state which authors and which funders?

The corresponding author received a salary from the funders.

2d. if you did not receive any funding for this study, please state: “The authors received no specific funding for this work.”

Please see the response under query 2c

3. Thank you for stating the following in your Competing Interests section: "No"

Competing Interests on the online submission form is completed as stated (The authors have declared that no competing interests exist).

5. In your Data Availability statement, you have not specified where the minimal data set underlying the results described in your manuscript can be found……………..

Now our underlining data set is uploaded as Supporting Information files

5. PLOS requires an ORCID iD for the corresponding author in Editorial Manager on papers submitted after December 6th, 2016…………………….

Thank you for the information. Now the ORCID ID of the corresponding author is updated

6. Please amend either the abstract on the online submission form (via Edit Submission) or the abstract in the manuscript so that they are identical.

We edited and made as the abstract at both site (online submission form and revised manuscript) are identical

7. We note that [Figure(s) 1] in your submission contain map images which may be copyrighted.

Thank you for your note, but the source of this map is the authors. We now sketched the map by including necessary map information (Fig 1) 

Now we included Supporting Information files at the end of manuscript by following PLOS ONE format and style guideline

Additional Editor Comments:

9. We noticed you have some minor occurrence of overlapping text with the following previous publication(s), which needs to be addressed…………………………. It seems thus that the data on a stock basis in the present manuscript may derive from the ones of the former article, thus invite you to revise the present ms and clearly indicate the differences between the article in the J. Degrade. Min. Land Manage and the present one. Also, please carefully take into account the reviewer’s suggestions.

Thank you for your suggestion. In this research, the gap which was not addressed in the previous research which is SOC and TN stocks were determined and compared. Therefore, this research can add a value for the scientific knowledge. Even if the sample data is collected from the same sampling unit, the objective of the first work (previously published) is different from this work. In this research, we estimate and compare the soil organic carbon and nitrogen stocks which are not addressed in the first activities. Due to this we confirmed in our previous cover letter as this work is original and has not been published elsewhere, nor is it currently under consideration for publication elsewhere. 

Lastly, I invite you to clearly diversify the previous article from the present ms. Wide parts of the materials and methods may be shared, but not of the other sections. In addition, data should not be replicated within the ms, so that just 1 table is enough.

We excuse for not clarifying it. We now made great efforts for clarification all parts of the manuscript 

In addition, tab. 4 is useless. You can just report the one number (R=0.86) in the main text.

Now, we removed table 4 and only report the value in the text

Reviewer #1: 

1. The paper is written in poor English. There are many grammar mistakes and some sentences that are not easily understood. A very careful and intensive revision by a native speaker is necessary.

We have endeavored to address all comments in this revised manuscript and believe the manuscript is now greatly improved, including the language. 

2. The title corresponds to the text. However, it has to be redone and reformulated.

Now we modified our title for differentiation and easily understandable 

3. The authors must pay attention to the abbreviation. Try to use the same abbreviation. Please specify if the values are for SOC or TN amount or for the stocks .

4. Thank you for your comment. Now, we have made great efforts to use abbreviations consistently throughout the revised manuscript. The value is for SOC and TN stocks

Abstract Section 

5. Line 22 : “The result indicated that farmland treated with LSB of 6 years has insignificantly higher SOC (102.57±9.00 Mg ha-1) and TN (9.74±0.9 Mg ha-1) than farmland treated with LSB of 3 years (98.42±10.24 Mg ha-1).” It is preferable that the authors add the value of TN for LSB of 3 years for a better comprehension 

We accepted the comment and now we add the value of TN for LSB of 3 years as suggested in the revised manuscript (Page 2, line 55-56). 

5. Line 26 :“Similarly, farmland treated with LSB of 6 years was sequestered more 5.20% % of SOC ..” Please delete the double symbol %

It is deleted from the abstract part of the revised manuscript as commented

Introduction section

6. The introduction should be developed. , there are many researches related to this problematic have been done. It would benefit of more detail and supported with the related references.

We have now considerably improved the introduction section as suggested in the revised manuscript

7. Line 48 :“In addition, important terrain like a slope, aspect and soil types...” Slope, aspect and soil types are considered like factors or parameters not terrain

We accepted the comment and replaced “, terrain” by “factors’ in the revised manuscript (page 3, line 85-86)

8. Line 58 “ Similarly, fields treated with soil and water conservation practices were founded the lower mean bulk density than the untreated fields in Adaa Berga district, western Ethiopia [15].” Please rephrase appropriately

We accepted the comment, and now clarified the sentence in the revised manuscript (page 4, line 95-96)

Materials and methods section

Description of the study area, line 86 :“ 7°46’’00’N-7°47’’00’N latitude and 36°44’’10’E-36°46’’50’E longitude (Fig 1).” These coordinates system does not correspond to the Fig 1.

We now sketched the coordinate system explicitly for the study area as our description match with the text (Fig 1)

10. Figure 2 : On the axis of temperature please add the unit and delete “Max. And Min.” because it is defined on the legend.

We deleted and added the unit as commented ( Fig 2)

11. Research design and soil sampling, line 103-106 and line 107-112 : Please rephrase appropriately to avoid redundancy

We have now considerably improved the sentences (Page 5, Line 139-142)

12. Research design and soil sampling, line 115 “Finally, SOC (Mg ha-1) and TN (Mg ha-1) were calculated….” Please specify that it is SOC and TN stocks.

We clarified it as commented in the revised manuscript (page 6, line 152-154)

13. Research design and soil sampling, line 112-117 “The collected soil samples were transported to Jimma Agricultural Research Centre laboratory … bulk density (g/cm3), and layer thickness (cm).” The authors should move this paragraph in the laboratory analysis section.

We now moved the mentioned paragraph as commented in the revised manuscript (page 6, line 147-149)

14. The authors must improve the table 1, 2 and 3 : Specify that the values are for SOC and TN stocks as l mentioned before. Also, please try to be brief when writing the name of x-axis, it could be SOC stock LSB-3 years / LSB-6years/ Control.

Now, we improved the x-axis of merged table as commented in the revised manuscript (Table 1)

Results and discussion section

15. The authors showed that total nitrogen, organic carbon and bulk density were used in this paper, but in the results and discussion section, why it was not included? The authors should add these data to this section using a table

Thanks for your suggestion, but since the main objective of this paper is to determine and compare the SOC and TN of farmland treated with 3-years and 6-years and adjacent farmland without conservation measures we didn’t include organic carbon fraction, nitrogen and bulk density in the result part to avoid the redundancy. A reviewer of this paper can find that information from the supportive file folder. 

16. Line 149 : TOC and TN data can not completely represent soil organic matter content. As fields treated with above three years’ soil and water conservation measures had higher organic matter accumulation as compared to the non-conserved fields. The data of OM would be more accurate.

Thank you for your suggestion. In this research, the SOC and TN stocks of LSB treated and untreated farmland were determined and compared. Investigation on Soil organic carbon and total Nitrogen stocks is a fundamental to know: 

The fertility, Chemical, physical and biological properties of soil

Soil structure: The water holding capacity and rainfall infiltration properties of organic carbon soils creates better landscape moisture availability. Root development and rainfall variation tolerance is also significantly enhanced in soils with improved aggregation from carbon. Therefore, Investigation of SOC is the strong indicator of the soils biological health.

Ecological soil function: The type and function of soil microbes is impacted by the availability of soil organic carbon and Nitrogen stocks. A healthy soil system supports pasture nutrient uptake, assisting root growth and crop disease suppression has also been associated. So, determination of soil organic carbon and nitrogen stock is a crucial way of identifying the ecological function of one soil. 

Basis of sustainable agriculture: In addition to the atmospheric benefits of carbon and nitrogen stocked in soils, the ecology and function of agricultural systems are improved. Resulting physically cohesive soil resists soil losses by wind or water erosion. So, determination of soil organic carbon and nitrogen is the bases for characterizing of farmland and implementation of further intervention.

17. Figure 3: Please specify that the values are for SOC stocks, on the y-axis and for the legend. Same as I mentioned below, try to write briefly, instead of writing “farmland with three years LSB” The authors can just write LSB-3years and Control.

We accepted the comment, and replaced “farmland with three years LSB” by “LSB-3years” and control in the revised manuscript (Figure 3)

18. Line 202-204 :“The top layer accounted for 52% of the TN in the upper bund zone of farmland treated with six years aged LSB and 51% of the TN in adjacent upper zone of farmland without conservation measures.” Please provide explanation how the authors get these values?

Thank you for explanation question. Now, the authors removed these confused values in the revised manuscript. 

19. Line 228 “The upper, middle and lower bund zones of SOC stocks in farmland treated with six years LSB was approximately 10.5%, 10.3% and 10.3% higher than the upper, middle and lower bund zones of farmland treated with three years LSB, respectively.” Please explain how the authors get these values?

Thank you for explanation question. Now, the authors removed these confused values in the revised manuscript.

20. For table 4: The Pearson`s correlation matrix for soil organic carbon and total Nitrogen, is it for the stocks values?

Yes. It is for stock values

21. References: Please be sure that all the references cited in the manuscript are also included in the reference list and vice versa with matching spellings and dates and according to the journal format, "Instructions for Authors".

Thank you for comment, we assured that our manuscript reference list and citation are followed the journal format, "Instructions for Authors".

 

Reviewer #2: 

1. The results section is divided into three subsections, but due to a data repetition in each subsection please consider a possibility to construct the results section comparing all 3 treatments in one table, if it`s possible to present data of statistical analysis this way.

Thank you for your comment, now we are merged all three tables at one in the revised manuscript

6. Data Availability Statement in the manuscript PDF file is Yes, but there is no information where the data may be found.

Now our underlining data set is uploaded as Supporting Information files

7. Add information to Research design section about the soil you were studied: Was it Nitisol or Acrisol?, What is the clay content in it?, etc. Is a sampling site on adjacent farmland without conservation measures representative? Was it the same soil type? Which type of crops was grown on adjacent farmland?

We accepted the comments and now the dominant soil type of the area which is Nitisolis described in the revised manuscript. Similarly crop which is grown on the control is also described (page 4, line 128-129)

8. Consider adding initial data (bulk density and content of organic carbon and total nitrogen in the soil) to the results section. This will give additional information to understand the reasons for your findings.

Thanks for your suggestion, but since the main objective of this paper is to determine and compare the SOC and TN of farmland treated with 3-years and 6-years and adjacent farmland without conservation measures we didn’t include organic carbon fraction, nitrogen and bulk density in the result part to avoid the redundancy. A reviewer of this paper can find further information from the uploaded Supporting Information files. 

9. Laboratory analyses. Line 120: Walkley and black (use Upper case)

Now, we rewrote as commented in the revised manuscript (Page 5, Line 150-151)

10. P. 7 Line 172: Text is about TN stocks but the reference is to a figure about SOC stocks (Fig. 3). Delete this reference and insert Figure 3 after Line 154.

Thank you for the comment, now we put the table at its appropriate place in the revised manuscript as commented 

11. Conclusions. Line 261: Finally, it could be concluded that… Avoid “enormous amounts of SOC and TN” because as you`ve found the mean difference in SOC and TN stocks is not significant between 3-years aged LSB and non-treated farmland. To conclude “considerable role in mitigating climate change by sequestering corresponding greenhouse gases” research findings do not seem to be completely enough.

Now, we have made great efforts to conclude our result in the revised manuscript ((Page 10, Line 157-260)

12. Fig. 3: Delete “a” and “b” symbols in the figure?

We deleted symbols as commented ( Fig 3)

13. Fig. 3 and tables: Round off numbers to 2 decimal places

We made it as commented in the revised manuscript (Fig. 3, Table 1)

14. Consider language editing to make some sentences more easily understandable

We have made an effort to address all comments in this revised manuscript and believe the manuscript is now greatly improved, including the language. 

 

Reviewer #3

Dear Authors, unfortunately, the manuscript seems very similar to a previous research published form the same or almost the same Authors. I cannot discuss about the quality if a similar experiment is already published "https://jdmlm.ub.ac.id/index.php/jdmlm/article/view/751".

I would be pleased to be invited to revise other research of yours, the topic is sound and actual.

Please try to provide a systematic literature search before to do any experiment, it might be useful perform a more comprehensive search with a structured query in a citation and abstract database such as Scopus or Web of Knowledge, as it was carried out in Modelling of Soil Organic Carbon in the Mediterranean area: a systematic map November 2018Rendiconti Online Societa Geologica Italiana 46/2018 DOI: 10.3301/ROL.2018.68

I suggest to read also other systematic map applications to improve the state of the art writing style.

Thank you for your suggestion. In this research, the gap which was not addressed in the previous research which is SOC and TN stocks were determined and compared. Therefore, this research can add a value for the scientific knowledge. Investigation on Soil organic carbon and total Nitrogen stocks is a fundamental to know:

 The fertility, Chemical, physical and biological properties of soil

Soil structure: The water holding capacity and rainfall infiltration properties of organic carbon soils creates better landscape moisture availability. Root development and rainfall variation tolerance is also significantly enhanced in soils with improved aggregation from carbon. Therefore, Investigation of SOC is the strong indicator of the soils biological health.

Ecological soil function: The type and function of soil microbes is impacted by the availability of soil organic carbon and Nitrogen stocks. A healthy soil system supports pasture nutrient uptake, assisting root growth and crop disease suppression has also been associated. So, determination of soil organic carbon and nitrogen stock is a crucial way of identifying the ecological function of one soil. 

Basis of sustainable agriculture: In addition to the atmospheric benefits of carbon and nitrogen stocked in soils, the ecology and function of agricultural systems are improved. Resulting physically cohesive soil resists soil losses by wind or water erosion. So, determination of soil organic carbon and nitrogen is the bases for characterizing of farmland and implementation of further intervention.

---

## [Decision Letter · Decision Letter 1]

11 Apr 2021

PONE-D-20-35754R1

Comparison of soil organic carbon and total nitrogen stocks between farmland treated with three and six years level soil bund and adjacent farmland without conservation measure: in the case of southwestern Ethiopia

PLOS ONE

Dear Dr. Gemechu,

Thank you for submitting your manuscript to PLOS ONE. After careful consideration, we feel that it has merit but does not fully meet PLOS ONE’s publication criteria as it currently stands. Therefore, we invite you to submit a revised version of the manuscript that addresses the points raised during the review process.

If applicable, we recommend that you deposit your laboratory protocols in protocols.io to enhance the reproducibility of your results. Protocols.io assigns your protocol its own identifier (DOI) so that it can be cited independently in the future. For instructions see: http://journals.plos.org/plosone/s/submission-guidelines#loc-laboratory-protocols. Additionally, PLOS ONE offers an option for publishing peer-reviewed Lab Protocol articles, which describe protocols hosted on protocols.io. Read more information on sharing protocols at https://plos.org/protocols?utm_medium=editorial-emailutm_source=authorlettersutm_campaign=protocols.

We look forward to receiving your revised manuscript.

Kind regards,

Sergio Saia, Ph.D.

Academic Editor

PLOS ONE

Journal Requirements:

Reviewers' comments:

Reviewer's Responses to Questions

**Comments to the Author**

1. If the authors have adequately addressed your comments raised in a previous round of review and you feel that this manuscript is now acceptable for publication, you may indicate that here to bypass the “Comments to the Author” section, enter your conflict of interest statement in the “Confidential to Editor” section, and submit your "Accept" recommendation.

Reviewer #1: All comments have been addressed

Reviewer #2: All comments have been addressed

Reviewer #3: All comments have been addressed

2. Is the manuscript technically sound, and do the data support the conclusions?

Reviewer #1: Yes

Reviewer #2: Yes

Reviewer #3: Partly

3. Has the statistical analysis been performed appropriately and rigorously? 

Reviewer #1: Yes

Reviewer #2: Yes

Reviewer #3: I Don't Know

4. Have the authors made all data underlying the findings in their manuscript fully available?

Reviewer #1: Yes

Reviewer #2: Yes

Reviewer #3: Yes

5. Is the manuscript presented in an intelligible fashion and written in standard English?

Reviewer #1: Yes

Reviewer #2: Yes

Reviewer #3: Yes

6. Review Comments to the Author

Reviewer #1: Review report

Manuscript Number: PONE-D-20-35754 “Comparison of soil organic carbon and total nitrogen stocks between farmland treated with level soil bund of different age and adjacent farmland without conservation measure: in the case of southwestern Ethiopia”

The manuscript addresses a subject of current interest related to the effect of level soil bund (aged 3 and 6 years) on Soil organic carbon and total nitrogen Stocks compared to a control farmland. Overall, I do like this paper, but the novelty and potential for impact are somewhat limited. I here encourage the authors to revise the paper.

In overall I fell the study can be suitable for publication in PLOS ONE after minor revision. Some suggestions for further improvement are given below and other are reported in attached word file:

Abstract

Line24-25: the authors said ‘’With respect to the age of LSB, farmland treated with LSB-6 years accumulated more 2.83% SOC stock than control.”

If they are comparing the overall mean of LSB-6 year to control, normally the corrected value will be 5.83% not 2.83%. Please check it

Results and discussion section

Line 203-205 : the authors said that ‘’ The overall mean value of LSB-6 years has exhibited higher mean value of SOC at the lower bund zone (97.81±10.49 Mg ha-1) as compared to LSB-3 years (94.37±7.65 Mg ha-1) (Table 1).’’

Or LSB-6 year exhibited more SOC stock at the lower and middle bund zone.

Line 208-211: the authors said “TN socks did not show significant difference (p0.05) between all inter bund zones. However, under all bund zones, the TN stock showed higher overall mean value under LSB-6 years (9.44±1.57 Mg ha-1) than LSB-3 years (9.28±1.60 Mg ha-1) (Table 1). Similar results were reported by [26] in Zikre watershed, Adaa Berga district. “

But for LSB-6 year, the TN stock is 9.37±1.10 Mg ha-1 and for the LSB-3 year is 9.44±1.57 Mg ha-1. In this case TN stock showed higher mean under LSB-3 year as compared to LSB-6 year.

Looking forward for your positive consideration of these comments.

Regards,

Reviewer of the manuscript

Reviewer #2: Here some suggestion to improve the manuscript:

1) Lines 139-140 “inter-bund zone …” use the same terminology that you use in the Results section: (upper, middle and lower) instead of (loss, middle and deposition)

2) In line 144 “and the same size undisturbed soil samples were collected for bulk density determination” do not repeat 141-142 lines “Similar sample sizes of undisturbed soil were also collected for bulk density determination using core sampler”, just use “bulk density determination”

3) Line 154: bulk density (g/cm3) (use superscript)

4) In table 1 do not repeat “SOC stock of/ TN stock of”. Write it only once at the top line of the table and indicate below: “control, LSB-3 years, LSB-6 years”

5) Lines 259-260: “this contribute in mitigating climate change…” According to obtained results it’s better to conclude the deposition of carbon and nitrogen than their sequestering (from the atmosphere). So, I suggest to write “by preventing erosion-induced greenhouse gases emission into the atmosphere” instead of “by sequestering corresponding greenhouse gases”

Reviewer #3: Dear Authors

I read the revised version of the manuscript “Comparison of soil organic carbon and total nitrogen stocks between farmland treated with level soil bund of different age and adjacent farmland without conservation measure: in the case of southwestern Ethiopia” I have seen some improvement and I think the paper can be revised again and go ahead in the publishing process, but please amend all the things listed below and the other review request. Reviewers suggested many things to improved your manuscript much than I have done due to the similarity with your previous unacknowledged research about SOC % with the same experimental settings. The experimental setting is decent, the authors can bring more evidences about the study area and they can translate the effect found in the trial in order to give a practical benefit addressed to the local policy makers.

The manuscript is proposed as a research article, due to the length (3200 words) the similarity with the previous study (shared experimental settings) of the same work of the authors about SOC %, and the limited novelty, I think it can be proposed as a short communication, a normal research article is made by approximately 6000 works, although I found normal manuscript of 8000 words.

Line24-25: Please check it ‘’With respect to the age of LSB, farmland treated with LSB-6 years accumulated more 2.83% SOC stock than control.” It should be around 5.83%?

Terminology must be consistent throughout the manuscript (e.g. lines 139-140 inter-bund zone), please check the consistency

I would rephrase the sentence “by sequestering corresponding greenhouse gases” in something more neutral.

Please define better the conclusion of the experiment. Do not oversell the findings.

Why not include the authors who revised the manuscript? They must be capable to understand the problem to check the language. I do not know in what extend they were involved in the research activities but in case they were, they must be rewarded.

Kind regards

7. PLOS authors have the option to publish the peer review history of their article (what does this mean?). If published, this will include your full peer review and any attached files.

Reviewer #1: No

Reviewer #2: No

Reviewer #3: **Yes: **Calogero Schillaci

---

## [Author Response · Author response to Decision Letter 1]

27 Apr 2021

Attached as supportive materials

---

## [Editor Report · Decision Letter 2]

11 May 2021

Comparison of soil organic carbon and total nitrogen stocks between farmland treated with three and six years level soil bund and adjacent farmland without conservation measure: in the case of southwestern Ethiopia

PONE-D-20-35754R2

Dear Dr. Gemechu,

We’re pleased to inform you that your manuscript has been judged scientifically suitable for publication and will be formally accepted for publication once it meets all outstanding technical requirements.

Kind regards,

Sergio Saia, Ph.D.

Academic Editor

PLOS ONE
---

## [Editor Report · Acceptance letter]

17 May 2021

PONE-D-20-35754R2 

Comparison of soil organic carbon and total nitrogen stocks between farmland treated with three and six years level soil bund and adjacent farmland without conservation measure: in the case of southwestern Ethiopia 

Dear Dr. Hailu:

I'm pleased to inform you that your manuscript has been deemed suitable for publication in PLOS ONE. Congratulations! Your manuscript is now with our production department. 

Kind regards, 

on behalf of

prof Sergio Saia 

Academic Editor

PLOS ONE